# Drivers of the Adoption and Exclusive Use of Clean Fuel for Cooking in Sub-Saharan Africa: Learnings and Policy Considerations from Cameroon

**DOI:** 10.3390/ijerph17165874

**Published:** 2020-08-13

**Authors:** Alison Pye, Sara Ronzi, Bertrand Hugo Mbatchou Ngahane, Elisa Puzzolo, Atongno Humphrey Ashu, Daniel Pope

**Affiliations:** 1Public Health England North West, Preston PR1 0LD, UK; alison.pye1@nhs.net; 2Department of Public Health and Policy and Systems, University of Liverpool, Liverpool L69 3GB, UK; Sara.Ronzi@lshtm.ac.uk (S.R.); elisa.puzzolo@glpgp.org (E.P.); 3Department of Health Services Research and Policy, Faculty of Public Health & Policy, London School of Hygiene and Tropical Medicine, London WC1H 9SH, UK; 4Douala General Hospital, Douala 3554, Cameroon; mbatchou.ngahane@yahoo.com (B.H.M.N.); ashu2002200@yahoo.com (A.H.A.); 5Global LPG Partnership, New York, NY 10065, USA

**Keywords:** household air pollution, clean cooking, clean fuel, LPG, Cameroon, Sub-Saharan Africa

## Abstract

Household air pollution (HAP) caused by the combustion of solid fuels for cooking and heating is responsible for almost 5% of the global burden of disease. In response, the World Health Organisation (WHO) has recommended the urgent need to scale the adoption of clean fuels, such as liquefied petroleum gas (LPG), in low and middle-income countries (LMICs). To understand the drivers of the adoption and exclusive use of LPG for cooking, we analysed representative survey data from 3343 peri-urban and rural households in Southwest Cameroon. Surveys used standardised tools to collect information on fuel use, socio-demographic and household characteristics and use of LPG for clean cooking. Most households reported LPG to be clean (95%) and efficient (88%), but many also perceived it to be expensive (69%) and unsafe (64%). Positive perceptions about LPG’s safety (OR = 2.49, 95% CI = 2.04, 3.05), cooking speed (OR = 4.31, 95% CI = 2.62, 7.10), affordability (OR = 1.7, 95% CI = 1.38, 2.09), availability (OR = 2.17, 95% CI = 1.72, 2.73), and its ability to cook most dishes (OR = 3.79, 95% CI = 2.87, 5.01), were significantly associated with exclusive LPG use. Socio-economic status (higher education) and household wealth (higher income) were also associated with a greater likelihood of LPG adoption. Effective strategies to raise awareness around safe use of LPG and interventions to address financial barriers are needed to scale wider adoption and sustained use of LPG for clean cooking, displacing reliance on polluting solid fuels.

## 1. Introduction

It is estimated that over 700 million people living in Sub-Saharan Africa (SSA) are dependent on polluting biomass fuels (such as wood and charcoal) for household cooking [1]. These are often burnt on inefficient open fires or traditional stoves, resulting in high levels of air pollutants, such as respirable fine particulate matter (PM_2.5_) and carbon monoxide (CO), which pose a major risk to health [2]. This household air pollution (HAP) is now understood to be causally related to chronic lung disease, lung cancer, ischaemic heart disease and stroke in adults, and acute lower respiratory infections in children [3]. In 2017, HAP from solid fuel use was estimated to account for over 390,000 deaths (5.3% of all deaths) and 4.1% of disability adjusted life years (DALYs) in SSA [4]. The World Health Organisation (WHO) recommends that PM_2.5_ levels should ideally be reduced to less than an annual mean of 10 µg/m^3^ to significantly reduce or eliminate the negative health impacts of HAP [2]. However, a more practical interim target level (IT-1) of 35 µg/m^3^ has been set for countries where baseline PM_2.5_ levels are high or very high [2].

To achieve those targets, much interest arose in burning biomass more cleanly using cookstoves with improved combustion and/or venting through chimneys. This was summarised in a systematic review and meta-analysis (SRMA) of real-life effectiveness of “improved” stoves and clean fuels in reducing HAP [5]. However, as reported in the SRMA, most of these interventions do not get close to achieving WHO IT-1 levels of PM_2.5_ or to reducing the harms associated with HAP [5,6,7,8]. Accordingly, the focus has now shifted to supporting the increased use of clean, modern fuels such as liquified petroleum gas (LPG) and electricity in order to achieve the United Nations’ Sustainable Development Goal (SDG) 7 on universal energy access by 2030 [9,10,11]. The effectiveness of such fuels in having positive impacts on health is dependent on their sustained and exclusive use. Evidence suggests that in many low and middle-income countries (LMICs), traditional polluting fuels are often used concurrently with cleaner fuels (this issue is known as “fuel-stacking”), meaning that many of the health benefits are lost [3]. 

In addition to the health impacts of HAP, there are important implications for the climate, with emissions from solid fuel combustion, including black carbon and greenhouse gases, contributing to global warming [12]. Reliance on wood and charcoal is also a major contributor to environmental degradation and deforestation which indirectly contribute to climate warming and habitat reductions [12]. In SSA, it is estimated that 70% of current deforestation may be related to the demand for wood and charcoal for household energy [13]. Environmental concerns are major policy drivers for scaling clean fuels in many LMICs. A number of countries in SSA have developed national plans to increase uptake of LPG for domestic use amongst their populations under the Sustainable Energy for All (SE4All) initiative and more recently SGD 7 [14,15,16]. LPG is an ideal clean cooking fuel to promote in many countries because it is portable; easy to store and transport; and requires much less costly infrastructure compared to natural gas or electricity, meaning that adoption can be rapidly scaled [14,17]. It is also a safe fuel (when used and regulated correctly) and provides substantial health and environmental benefits [11,14,18,19]. In its 2017 report, the International Energy Agency (IEA) projected LPG to be an achievable clean cooking solution for 1.4 billion people by 2030 [10]. In 2016, the Government of Cameroon published a national LPG Masterplan outlining their commitment and detailed strategy to expand LPG use from 12% in 2012 to 58% of households by 2035, largely by improving its availability and accessibility through investment in cylinders and infrastructure [20]. If achieved, it is estimated that 23,000 lives could be saved, and 760,000 DALYs averted by 2030 [21]. In addition, there will be associated climate co-benefits leading to a global cooling of −0.1 milli °C in 2030 [21]. 

To inform national policies concerned with scaling LPG in Cameroon, the LPG Adoption in Cameroon Evaluation (LACE) studies were launched in 2016, coinciding with the announcement of the national LPG Master Plan. The studies aimed to identify barriers and enablers to using LPG as a household fuel and to test community-based interventions to address these barriers. A publication of findings from the LACE surveys of two communities in Southwest Cameroon found that education, household wealth and socio-economic status were all predictors of LPG adoption and exclusive use [22]. A further qualitative participatory study (using visual photovoice methods) found that affordability and safety concerns were key barriers to LPG adoption, whilst an awareness of the health benefits and increased availability in rural areas were enabling factors [23]. Over the last 30 years, a number of studies have investigated the determinants of uptake and sustained use of clean cooking fuels and technologies in LMICs. A consistent finding from this work is that socio-economic factors such as educational attainment and wealth are important in transitioning to clean cooking [24,25,26,27,28,29,30,31]. The latter is in accordance with the “energy ladder” hypothesis, which suggests that with rising affluence, households will transition from polluting fuels to using cleaner, more modern ones. Another consistent observation is that once clean fuels/technologies are adopted for cooking, they are rarely used exclusively. Instead, polluting fuels, such as wood and charcoal, are often used concurrently (so-called “fuel stacking”) [32,33,34,35,36,37]. The relative costs of domestic fuels (including initial stove acquisition and ongoing fuel costs) are also important predictors of their uptake and sustained use [35,38,39], having a greater impact for low income households [38] and in areas where solid fuels can be collected for free [35]. An additional barrier to the adoption of cleaner cooking is the lack of access to or availability of clean fuels, with urban households having greater access to and more adoption of clean fuels compared to rural households [24,26,40,41,42]. Positive perceptions of cooking with clean fuels relative to solid fuels also affect their uptake and sustained use, with taste, safety and the ability of a fuel to fulfil household cooking requirements influencing choice [27,43,44,45,46,47,48]. 

The current study aims to quantitatively summarise socio-economic and individual drivers of LPG adoption and sustained use as a clean cooking fuel in Southwest Cameroon. It builds on the results of the previous mixed-methods research in the region [22,23] by utilising a larger dataset on fuel use, household and individual factors and fuel preferences. The large sample size enables a more robust analysis of predictors of (i) any and (ii) sustained/exclusive use of LPG. Crucially, perception data also offers important insights into opinions of LPG, and the impacts of these perceptions on fuel choice. The results are highly relevant to Cameroon stakeholders involved in developing policies alongside the implementation of the National LPG Masterplan to achieve aspirational targets of adoption. 

## 2. Materials and Methods 

### 2.1. Questionnaire Development

Questionnaires for the quantitative surveys were developed using existing pre-validated questions (e.g., WHO questions set for fuel use) and new questions derived through expert consultation and subsequent piloting (e.g., fuel preference questions). The surveys included information on the demographic composition of each household, detailed primary and secondary fuel use, cooking behaviours and each respondent’s perceptions of LPG as a household fuel. These perceptions included speed, cleanliness, availability, affordability, safety, its ability to cook dishes and the ease of replacing cylinders, each of which were rated on a four-point scale (ranging from very good to very bad) (Table 1). 

### 2.2. Participants

The study population consisted of three peri-urban communities and one rural community in the anglophone region of Southwest Cameroon (total 3343 households). The surveys were conducted in two phases during the LACE studies. Phase 1 (LACE-1) was conducted in 2016 and involved 1577 households from two health districts in the Southwest Region of Cameroon (peri-urban Limbe and rural Buea). Initial descriptive findings relating to characteristics of the survey population and fuel use have been published by Pope et al. [22]. Phase 2 (LACE-2) was conducted in 2017 and involved households being surveyed from two additional peri-urban communities in Limbe (Batoke (*n* = 707) and Botaland (*n* = 1059)) as part of an evaluation of microfinance to support LPG adoption. Further details of these two phases are shown in Table 2.

### 2.3. Data Collection

All the surveys were conducted by eight trained fieldworkers once the communities had been sensitised and enumerated. For each household in both phases, the main cook and/or household head was asked to complete the questionnaire (due to their direct involvement with the usage and purchase of household fuel). These were administered in English, or when appropriate, in the local dialect (Pidgin). Written informed consent was obtained from participants at the point of questionnaire delivery, and data were anonymised at collection. Ethical approval was granted by the Cameroonian ethics committee (Comité National D’Ethique de la Recherche pour la Santé Humaine), and subsequently from the University of Liverpool’s ethics committee.

### 2.4. Statistical Analysis

Primary use of LPG fuel (compared to wood/charcoal/sawdust/kerosene) was the main outcome of interest. Potential predictors of LPG use included the following variables: (i) socio-demographic, wealth and individual factors (age, sex, education, marital status, income, asset ownership, transport), (ii) household characteristics (home ownership, people resident in home, number of rooms) and (iii) LPG perceptions (speed, safety, ability to cook most dishes, affordability, cleanliness, ease of replacing cylinders and cylinder availability). A descriptive comparison of LPG use by the predictor variables was summarised using appropriate hypothesis testing: categorical (chi-squared tests) and continuous (Kruskall Wallis–Mann Whitney U) comparisons—a statistical significance value of 5% was used, with a Bonferroni correction being applied for multiple comparisons. Unconditional logistic regression was used to summarise the relationship between LPG use (dependent variable) and socio-economic, wealth, individual and household characteristics and LPG fuel perceptions (independent variables). Associations were summarised through odds ratios and 95% confidence intervals. The outcome (LPG use) was classified as “any LPG use” (households using LPG as either their primary or secondary cooking fuel) and “exclusive LPG use” (households reporting using LPG as their only cooking fuel). To identify factors most strongly associated with LPG use, multivariable logistic regression analysis was conducted, including all factors found to be significantly associated with LPG use in univariate analysis. To summarise the predictive strength of the final model, a Hosmer–Lemeshow goodness of fit test was conducted. All analyses were conducted using Stata v14 (StataCorp LLC, College Station, TX, USA) [49].

## 3. Results

### 3.1. General Characteristics of the Household and Household Head

Survey data were obtained from a total of 3343 households; 243 rural households from Buea and 3100 peri-urban households from Limbe (Mile 4, Botaland and Batoke). Response rates were high (>90%) when households were resident at the time of visit, possibly resulting from the process of sensitising communities, facilitated by community chiefs and town criers. In rural Buea, only approximately half the available homes were surveyed due to the difficulty of finding people at home during the survey period (households typically worked in agriculture/farming during the day). 

The characteristics of the participants who took part in the surveys are shown in Table 3. Just over half had a female household head (*n* = 1727; 52%), median age 36 years. The majority reported being married or in a partnership (*n* = 1916; 57%) and having completed primary (*n* = 3234; 97%) or secondary (*n* = 2061; 62%) education. In peri-urban areas, people typically rented their houses (*n* = 1591; 51%), whereas in the rural community, home ownership was more common (*n* = 213; 88%). Almost two thirds of homes did not have a flushable toilet (*n* = 2019; 60%) or access to piped water (*n* = 1973; 59%), even though approximately half (*n* = 1564; 58%) of household incomes exceeded the national average of 50,000 CFA (Central African Francs) (83 United States Dollars (USD)) per month (all currency conversions correct as of 3rd September 2019).

Compared to peri-urban household heads, rural household heads were more likely to be male (71% vs. 47%; *p* < 0.0005), older (median age 52 vs. 38 years; *p* < 0.0005) and single (31% vs. 15%; *p* < 0.0005). They were also less likely to have received a secondary education (36% vs. 63%; *p* < 0.0005). Rural households also typically had lower levels of income and asset ownership compared to peri-urban households. They were less likely to earn above the poverty threshold of 25,000 CFA per month (42 USD; 74% vs. 93% respectively; *p* < 0.0005); were less likely to be paid in cash (20% vs. 5%; *p* < 0.0005); and were less likely to own a flushable toilet (16% vs. 42%; *p* < 0.0005), have piped water (31% vs. 42%; *p* < 0.0005) and have access to a car (18% vs. 2%; *p* < 0.0005).

### 3.2. Patterns of Fuel Use

Reported primary and secondary fuels used for cooking are shown in Table 4. The two main primary fuels were LPG (*n* = 1883; 56%) and wood (*n* = 1205; 36%), but other primary fuels included kerosene (*n* = 114; 3.4%), charcoal (*n* = 51; 1.5%), sawdust (*n* = 53; 1.6%), piped gas (*n* = 1; 0.03%) and electricity (*n* = 11; 0.3%). As one would expect, rural homes were significantly more likely to use wood as their primary cooking fuel (81% vs. 33%, *p* < 0.0005) and less likely to use LPG (16% vs. 66%%, *p* < 0.0005) given the availability of freely gathered wood in the rural community. 

Most households (70%) used multiple fuels for cooking (fuel stacking). The most common fuel combinations were LPG and wood (57%), LPG and charcoal (18%) and kerosene and wood (10%). Amongst households using LPG as either a primary or secondary cooking fuel, the majority (81%) exhibited fuel stacking behaviour. A similar pattern was seen across all four communities (see Figure 1).

### 3.3. Perceptions of LPG as a Cooking Fuel

Perceptions of LPG as a cooking fuel are shown in Table 5. The majority of participants thought that LPG was clean (*n* = 2808; 95%), obtainable (*n* = 1721; 62%), cooked most foods (*n* = 1909; 66%) and did so quickly (*n* = 2542; 88%). However, a high proportion of respondents reported concerns over the safety of LPG. Overall, more than two thirds of people (*n* = 1917; 64%) reported thinking that the fuel was either dangerous or very dangerous. These concerns could present a significant barrier to LPG adoption, although interestingly, a large proportion of LPG users (*n* = 1274/1937; 66%) and exclusive LPG users (*n* = 221/471; 45%) also reported thinking that it was unsafe while continuing to use it regularly. In addition to concerns about safety, many respondents expressed negative views relating to the cost of LPG. The majority of people reported that LPG refills were either expensive or very expensive (*n* = 1958; 69%). This view was most apparent amongst those who reported that they did not use LPG (*n* = 339/462; 73%), although it was also common amongst any LPG users (*n* = 1346/1900; 71%) and exclusive LPG users (*n* = 273/464; 59%).

Perceptions of LPG differed significantly by socio-demographic group (Table 6). In general (but not always), younger, wealthier, more educated household heads living in peri-urban areas were more likely to view LPG favourably. 

To investigate how perceptions of LPG as a cooking fuel might be related to adoption and exclusive use, opinions of LPG in terms of its efficiency, safety, cleanliness, availability, cost and ability to cook most dishes were stratified according to LPG use (Table 5 and Figure 2). Perceptions of LPG across all categories were associated with the degree of LPG use, such that respondents who reported positive views about LPG were significantly more likely to be using it. Furthermore, the proportion of people expressing positive opinions about LPG in each category increased according to increasing levels of adoption (Figure 2). This meant that those who reported using LPG exclusively were the most likely to express positive views about it (although even in this group, it was apparent that some people still had negative perceptions).

Respondents who perceived LPG to be safe were almost twice as likely to report using LPG (either alone or alongside other fuels), compared to those who reporting thinking it was dangerous (OR = 1.63, 95% CI = 1.31, 2.03) (Table 7). Similarly, respondents who stated that LPG could be used to cook most dishes were significantly more likely to report using it (OR = 1.60, 95% CI = 1.29, 1.9). However, perceptions that LPG cooked food quickly, was clean, was affordable and was available, and that the cylinders were easy to replace were not significantly associated with LPG use. 

Perceptions of LPG appeared to play a more important role in determining whether respondents used LPG exclusively compared to whether they used it at all. Positive perceptions of LPG in all categories (apart from cleanliness) significantly increased the likelihood of exclusive LPG use (Table 7). Respondents who believed that LPG cooked food quickly were four times more likely to report using it exclusively compared to those who thought it cooked food slowly (OR = 4.31, 95% CI = 2.62, 7.10). Similarly, respondents were significantly more likely to report having adopted LPG exclusively if they viewed it as safe (OR = 2.49, 95% CI = 2.04, 3.05), affordable (OR = 1.7, 95% CI = 1.38, 2.09), available (OR = 2.17, 95% CI = 1.72, 2.73) or able to cook most dishes (OR = 3.79, 95% CI = 2.87, 5.01), or thought that cylinders were easy to replace (OR = 1.87, 95% CI = 1.50, 2.34).

### 3.4. Socio-Economic Factors Affecting Fuel Use in Peri-Urban Households

Associations between socio-economic factors and LPG adoption or exclusive use were assessed in peri-urban households (Table 8). Older household heads (36–60 years) were significantly less likely to report using any LPG (OR = 0.64, 95% CI = 0.54, 0.77) or to report using it exclusively (OR = 0.30, 95% CI = 0.24, 0.38) compared to their younger counterparts (<36 years), as were household heads over the age of 60 years (OR = 0.30, 95% CI = 0.22, 0.41 and OR = 0.14, 95% CI = 0.07, 0.28 respectively). Household heads who had completed a secondary school (OR = 8.36, 95% CI = 5.41, 12.92) or university education (OR = 21.0, 95% CI = 12.52, 35.23) were significantly more likely to report using any LPG, and to report using it exclusively (OR = 10.21; 95% CI = 2.50, 41.71 and OR = 13.84; 95% CI = 3.36, 56.92 respectively) compared to those reporting no education. Those who had received a primary education only were also more likely to report using LPG than those with no education (OR = 2.64, 95% CI = 1.72, 4.07), but were no more likely to report using it exclusively.

Reporting of any use of LPG significantly increased with rising income (Table 8). Those earning over 301,000 CFA (502 USD) per month were nearly 18 times more likely to report using LPG compared to those earning less than the national average income of 50,000 CFA (83 USD; OR = 17.73, CI = 4.32, 72.85). Other measures of increased socio-economic status such as ownership of a flushable toilet (OR = 5.52, 95% CI = 4.45, 6.85), piped water (OR = 3.71, 95% CI = 3.05, 4.51) and access to a car (OR = 2.34, 95% CI = 1.91, 2.86) were also associated with any LPG use, whilst access to a motorbike or truck were not. No relationship was seen between income (or other wealth indicators) and exclusive LPG use, although respondents who were paid for their work in kind or not at all were significantly less likely to use LPG exclusively compared to those paid entirely in cash (OR = 0.40, 95% CI = 0.30, 0.53).

Increasing family size had no effect on any LPG use, but was significantly associated with exclusive use such that peri-urban households containing more than six people were 92% less likely to use LPG exclusively compared to those with 1–3 people (OR = 0.08, 95% CI = 0.05, 0.13). Similarly, although livestock ownership was not related to any LPG use, households who owned their own livestock were 87% less likely to use it exclusively (OR = 0.13, 95% CI = 0.06, 0.27) compared to those without.

Two multivariable models were developed for each outcome; (i) any use of LPG and (ii) exclusive use of LPG, based on the sets of covariates significantly associated with each outcome in univariate analysis (Table 9). For “any use of LPG,” younger age, education, increasing income up to 300,000 CFA per month (500 USD), ownership of a flushable toilet, having piped water, a cash income, access to a car and cooking inside were independent predictors (Hosmer–Lemeshow test *p* = 0.3811; R-square = 0.343). For “exclusive use of LPG,” younger age, being single, larger household size (number of people resident), overcrowding, a cash income, owning livestock and cooking indoors were all independent predictors (Hosmer–Lemeshow test *p* = 0.8128; R-square = 0.236).

Table 10 provides a summary of the identified enablers and barriers that were significantly associated with LPG adoption and/or exclusive use.

## 4. Discussion

### 4.1. Fuel Use Patterns and Their Determinants

This study utilises a unique dataset comprising results from over 3300 peri-urban and rural households in the Southwest Region of Cameroon. We explored perceptions of LPG as a clean cooking fuel and identified factors affecting its adoption and exclusive use. The findings are relevant for programs designed to scale widespread adoption of clean household energy.

We found that there is widespread use of LPG as a primary cooking fuel (56%), although mainly within peri-urban settings. In the rural context, only 16% of households (*n* = 38) reported using LPG as a primary cooking fuel with most (81%) using freely gathered wood. The wider use of LPG for clean cooking in peri-urban/urban compared to rural LMIC settings has been documented in the literature [24,26,40,41,42]. Reported reasons for this disparity include greater access to clean fuels in urban areas, lack of access to LPG in rural contexts, relative poverty and the opportunity costs of freely gathered biomass in rural settings [17,23,50]. In support of these hypotheses, this study found that people living in rural areas were significantly more likely to report perceiving LPG as expensive and more difficult to obtain than peri-urban households. Similarly, in their photovoice study, Ronzi et al. found that non-LPG users in the same study region reported difficulties obtaining LPG in rural areas due to the scarcity of retail shops and uncertain supply [23]. Understanding how to encourage and support adoption of LPG for clean cooking to address the negative impacts of solid fuel use on health (household air pollution) and the environment (deforestation) is challenging in rural contexts. Improving access to the LPG and providing financial support (e.g., through microfinance/subsidy) in these settings could help address barriers to the use of LPG for clean cooking, particularly if biomass can be collected for free.

Similar to other studies, we found that most households (70%) used more than one fuel for cooking. Of those households reporting use of LPG for cooking (as either a primary or secondary cooking fuel), four-fifths (81%) used it alongside a polluting solid fuel (wood or charcoal). It is well established that this “fuel stacking” is the norm for many households across SSA, Asia and Latin America [34,35,51,52,53,54]. The benefits of clean cooking with LPG are clearly reduced with a greater amount of fuel stacking with solid fuels and kerosene. A variety of reasons for fuel stacking have been described. These include varying fuel prices and inconsistent supply, perception of fuel “wastage” for foods with a long cooking duration, perceptions of taste and cultural tradition [3,32,48,54,55,56,57,58]. Addressing these barriers to encourage more exclusive use of LPG for clean cooking is important to maximising the health gains achieved from transitioning away from polluting solid fuels.

The socio-economic determinants of new adoption of clean household energy are well documented [24,25,26,27,28,29,30,31,59]. In accordance with this evidence, we found that increased education, wealth (income and assets) and younger age were all significantly associated with increased likelihood of LPG adoption, whereas larger household size (number of people resident) and land ownership were barriers to more exclusive use. In addition, we found that households reporting refills to be inexpensive were almost twice as likely to use LPG exclusively compared to those who did not. Qualitative studies conducted in Peru [43,45] and Mozambique [60], and in this Cameroonian setting [23,58], have reported perceived affordability as a crucial barrier to the adoption and continuous use of clean cooking fuels. Our findings are in accordance with the hypothesis that reducing the cost of LPG (e.g., through subsidy) could significantly increase its uptake among poorer households [44,61]. Evidence from Latin America has suggested that targeted LPG subsidies for resource poor households can be an effective way to assist with the equitable updating of clean household energy in vulnerable groups [62,63]. To support new adoption of clean cooking with LPG, financial innovations such as micro-loan schemes (to help with start-up costs) [64,65,66,67] and pay-as-you-cook smart meter technologies/schemes (to help with recurrent fuel costs) [68] are receiving increasing attention in SSA contexts by directly addressing the established barrier of affordability.

The drivers for transition to clean household energy are complex. In their systematic review, Puzzolo et al. [25] identified seven domains relevant to adoption and sustained use of clean fuels and technologies for cooking, including (i) fuel and technology characteristics, (ii) programs and policy context, (iv) tax and subsidies, (v) market development, (vi) regulation and legislation, (vii) household and setting characteristics and (viii) knowledge and perceptions. Findings from this study provide important insights for the last of these domains. In accordance with qualitative literature in other LMICs [23,44,45,54,69], we found that whilst the majority of participants reported LPG for cooking was clean (95%) and cooked food quickly (88%), they also perceived it to be both unsafe (64%) and expensive (69%). Familiarity with the fuel lessened these negative perceptions, with households reporting exclusive use of LPG being more likely to express positive views about it (more so than mixed and non-LPG users). This included positive views about LPG being cheap, available, efficient and able to cook most dishes. However even among the more frequent users of LPG, a notable proportion reported concerns over the safety of the fuel (45% of exclusive users).

Other studies from a variety of countries and settings have found concerns expressed by households over the safety of LPG as a cooking fuel, including studies from Mozambique, Peru, Cameroon and Nigeria [23,44,45,46,54,60]. These concerns are typically expressed in the absence of any direct experience of safety issues in relation to the fuel [45,70]. Concerns expressed include fears of gas leaks, fire and explosions, often based on hearsay rather than personal experience [45,54,70]. In following up with participants from our study through qualitative enquiry, non-LPG users and mixed fuel users reported safety concerns about LPG if they lived in wooden houses (due to the fire risk), or if the quality of LPG equipment (cylinders, valves or stove) was poor [23]. The findings from the current analysis identify that these safety concerns are not only common (64% of respondents) but can also be an important barrier to initial adoption and exclusive use. We found that respondents who reported LPG to be safe were significantly more likely to report any LPG use (OR = 1.63) or exclusive LPG use (OR = 2.49). Alleviating concerns on the safety of LPG for cooking through education is clearly a priority for policy to effectively scale widespread adoption of the fuel in Cameroon. This approach has been shown to increase the likelihood of LPG adoption in other LMIC settings [27].

### 4.2. Strengths and Limitations

We are confident that the surveys are representative of our study population in Southwest Cameroon due to the large sample size and high response rate. However, the number of households surveyed in rural areas was much smaller than the number in peri-urban areas, making the results more applicable to peri-urban settings. In addition, the study region generally has lower levels of poverty, higher levels of education and a greater proportion of homes with access to piped water compared to the rest of the country. This greater affluence in the study’s context corresponds to a greater proportion of households using LPG than the national average due to wider access to the fuel and affordability.

One unique facet of the study was the enquiry about perceptions of LPG as a cooking fuel in addition to actual use of LPG as a primary or secondary fuel. This allowed a comparison of safety concerns (which were surprisingly high in all fuel-using groups) by amount of LPG use, with residual concerns over safety still being reported in a notable proportion of exclusive users.

The cross-sectional nature of the study means that it is not possible to comment on the temporality of findings, although reported associations between socio-demographic and economic characteristics with use of LPG for clean cooking are consistent with observations from the literature. Of added value from this study is that, whilst the evidence for the association of these characteristics with adoption (or new use) of LPG confirms the findings from other studies in LMIC settings, the observation for the impact of the factors on exclusive LPG use is relatively unique. Whilst policies by the Government of Cameroon in relation to the National LPG Masterplan will be focused on increasing adoption of LPG, the benefits of clean cooking through reductions in household air pollution will only be experienced if the fuel is used in a sustained way and almost exclusively. To support this, it is crucial that LPG supply is consistent and reliable, and that the cost implication of using LPG for foods with long cooking durations is addressed.

### 4.3. Implications for Policy, Practice and Research

The LACE studies were developed alongside publication of the Cameroon National LPG Masterplan by the Ministry of Energy and Water (MINEE) to provide an evidence base for policies facilitating community transition from traditional use of polluting solid fuels to clean cooking with LPG. Results from this cross-sectional evaluation highlight some of the barriers to both adoption and sustained use of LPG for cooking that are amenable to strategic decision making for policy development. These include:
1.Addressing concerns around safety through effective messaging and education around LPG as a safe household fuel, including demonstration of its correct use to new users and careful inspection of LPG equipment;2.Addressing prohibitive costs of switching to LPG as a new cooking fuel through provision of financial assistance (microloans) to acquire the LPG equipment;3.Addressing concerns over the affordability of refills through maintenance of the fixed LPG subsidy (currently set by the Cameroon government at 6500 CFA (11 USD) for a 12.5 kg refill) and transport price equalization to regulate consistent pricing of LPG fuel in more remote areas.

The results for this cross-sectional analysis have identified a complex range of factors that influence both the adoption and exclusive use of LPG and highlight the need for a multi-faceted approach to policy in scaling its use for household energy [25]. A “whole system approach” to policy is required, encompassing both upstream and downstream interventions (e.g., from a community perspective) (Figure 3). One of the crucial barriers, highlighted above, is the prohibitive cost of becoming a new user for many low-income earners. Promotion of microloans or other financial incentives [17,66,71] to address upfront LPG equipment costs, and recurrent refill costs, is now receiving greater attention and community microfinance receives specific attention in the Cameroon National LPG Masterplan [20].

## 5. Conclusions

This study has used extensive cross-sectional survey data from rural and peri-urban communities in the Southwest Region of Cameroon to explore current fuel use practices and identify factors that affect both adoption and more exclusive use of LPG for clean cooking. In Southwest Cameroon it is evident that use of LPG as a primary cooking fuel is widespread (although largely restricted to peri-urban communities), reflecting the relatively mature LPG market in the region. Despite this, exclusive cooking with LPG was found to be rare, with concurrent use of other fuels (such as wood and charcoal) being relatively common. Predictors of adoption and sustained use include a higher education, rising income and younger age, whereas rural location, problems with availability, increased fuel costs and larger household size (increasing number of residents) appear to inhibit use of LPG.

The positive attributes of LPG were clearly expressed by study participants, with the majority reporting LPG as a convenient, clean and effective cooking fuel; however, a significant proportion of both users and non-users reported the fuel to be both unsafe and expensive. Familiarity with the fuel lessened these negative perceptions, with households reporting exclusive use of LPG being more likely to express positive views about it (compared to mixed and non-LPG users). Positive perceptions about LPG were particularly important for increasing the likelihood of exclusive use. Negative perceptions of LPG in terms of safety and cost will reduce the effectiveness of policies to effectively scale adoption and sustained use of the fuel for cooking. LPG is an inherently safe fuel when used correctly in a well-regulated market, wherein marketers take legal responsibility for maintaining the safety of LPG supply and cylinders (such as that in Cameroon). Education on safe cooking with LPG is likely to be an effective strategy to accompany other policies in order to scale adoption of the fuel in Cameroon to aspirational target levels (in line with the National LPG Masterplan) [20]. The issue of affordability is more complex. The perception of the increased cost of LPG relative to other purchased fuels is often misplaced and greater clarity on competing prices with charcoal and purchased wood might also be part of an education strategy. Direct intervention on affordability through microfinance (for example, supporting acquisition of LPG equipment through short term loans) is of interest to the Cameroon government (mentioned in the National LPG Masterplan) and has been extensively piloted in the country (Bottled Gas for a Better Life Initiative) [66,73]. This could well increase the adoption of LPG for resource-poor households in the country. Policy interventions such as value-added tax (VAT) exemption on LPG equipment or fuel bans on polluting fuels (e.g., on charcoal or kerosene) coupled with the existing LPG subsidies could stimulate more exclusive use of the fuel.

Beyond the national context for Cameroon, the results of this study are likely to be relevant to other Sub-Saharan African contexts, wherein there is an aspiration to scale adoption of LPG for clean household energy to achieve SDG 7 (universal energy access), while also contributing to reducing (i) the substantial burden of disease from household air pollution from combustion of solid fuels/kerosene and (ii) deforestation. 

## Figures and Tables

**Figure 1 ijerph-17-05874-f001:**
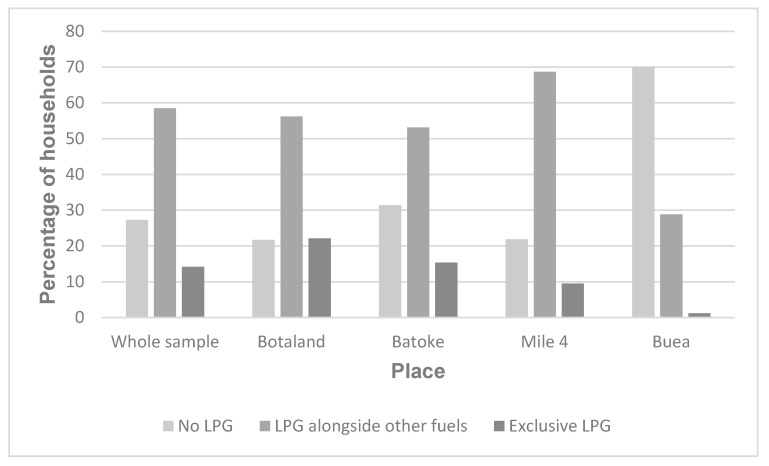
Reported degree of LPG use in the whole sample and by community (Buea is a rural community, and Botaland, Batoke and Mile 4 are all peri-urban communities).

**Figure 2 ijerph-17-05874-f002:**
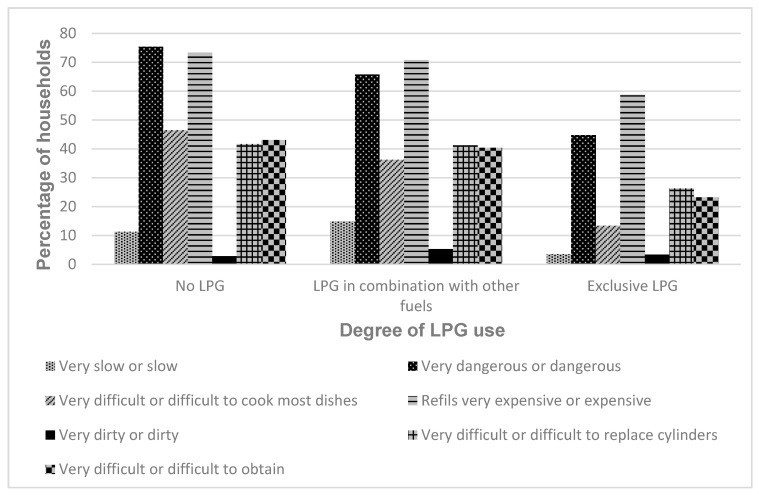
Perceptions of LPG according to whether it was used exclusively, in combination with another fuel or not at all.

**Figure 3 ijerph-17-05874-f003:**
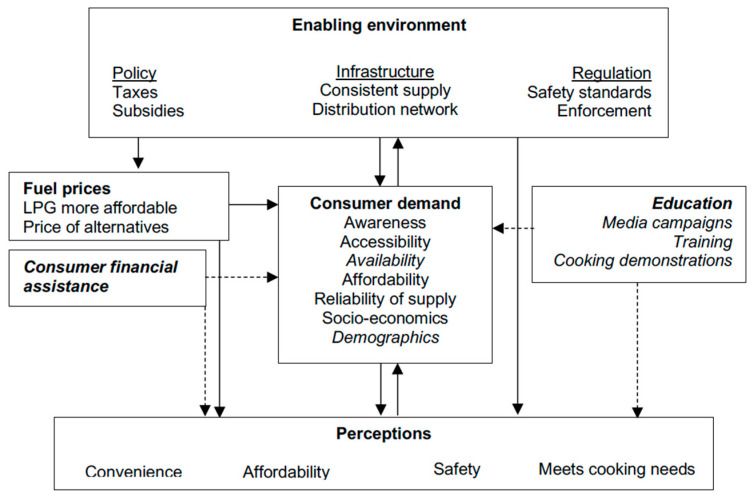
Interventions which would target some of the key factors effecting LPG uptake and exclusive use (adapted from Rosenthal et al. [72] (p. A6). New additions in italics).

**Table 1 ijerph-17-05874-t001:** Perceptions of LPG assessed through the questionnaires, with associated Likert scales.

LPG Attributes	Likert Scale
Speed of cooking	1 = Very slow 2 = Slow 3 = Fast 4 = Very fast
Ability to cook most dishes	1 = Very difficult 2 = Difficult 3 = It is okay 4 = Easy
Cleanliness (e.g., level of soot from the smoke)	1 = Very dirty 2 = Dirty 3 = Clean 4 = Very clean
Ease of replacing the cylinder (transporting to and from the shop)	1 = Very difficult 2 = Difficult 3 = It is okay 4 = Easy
Affordability of the refills	1 = Very expensive 2 = Expensive 3 = It is okay 4 = Cheap
Availability	1 = Very difficult to obtain 2 = Difficult to obtain 3 = It is okay 4 = Easy to obtain
Safety (e.g., fire or explosions)	1 = Very dangerous 2 = Dangerous 3 = Safe 4 = Very safe

**Table 2 ijerph-17-05874-t002:** Details of the designs, aims and sample sizes of the LACE-1 and LACE-2 studies.

Study	Main Aims	Survey Sample Size	Location	Recruitment	Design/Methods
LACE-1 (2016)	Describe fuels used for cooking. Identify factors affecting LPG adoption/persistent use. Assess how LPG adoption affects HAP, health and wellbeing.	1577	Mile 4 (peri-urban) Buea (rural)	Stratified random sampling All households eligible	Cross-sectional survey. Focus groups. Semi-structured interviews.
LACE-2 (2017)	Assess effectiveness of a micro-loan scheme. Assess effectiveness of the provision of a pressure cooker. Understand the social and cultural influences of LPG adoption/persistent use.	1766	Botaland (peri-urban) Batoke (peri-urban)	Simple random sampling	Before and after studies (150 households were provided with a micro-loan to help with LPG start-up costs, and 140 households were provided with a pressure cooker). Semi-structured interviews and focus group discussions. Photovoice participatory methods.

**Table 3 ijerph-17-05874-t003:** Characteristics of the households and household heads in the whole sample and stratified by rural and peri-urban communities.

Characteristic	Total Sample (*n* = 3343)		Peri-Urban (*n* = 3100)		Rural (*n* = 243)		*p* Value *
	No	%	No	%	No	%	
**Sex (household head)**							
Male	1616	48.3	1444	46.6	172	70.8	**<0.0005**
Female	1727	51.7	1656	53.4	71	29.2	
**Age (household head)**							
Median (IQR)	38.8 (14.3)		37.7 (13.6)		51.8(16.9)		**<0.0005**
18–25	543	16.3	530	17.1	13	5.4	
26–35	1128	33.8	1094	35.3	34	14.0	
36–45	771	23.1	722	23.3	49	20.2	**<0.0005**
46–65	734	22.0	640	20.7	94	38.7	
66+	166	5.0	113	3.7	53	21.8	
**Education (household head)**							
None	109	3.3	97	3.1	12	4.9	
Primary	1173	35.0	1032	33.3	141	58.0	**<0.0005**
Secondary	1471	44.0	1407	45.4	64	26.3	
University	590	17.7	564	18.2	26	10.7	
**Marital status (household head)**							
Married/partnership	1916	57.3	1804	58.2	112	46.1	
Widowed	726	21.7	686	22.1	40	16.5	**<0.0005**
Divorced/separated	152	4.6	137	4.42	15	6.2	
Single	549	16.4	473	15.3	76	31.3	
**People resident**							
Median (IQR)	4.7 (2.5)		4.7 (2.5)		4.6 (2.9)		0.2722
**Number of rooms** (excl. kitchen + store)							
Median (IQR)	2 (2)		2 (2)		4 (2)		**<0.0005**
**People per room**							
Median (IQR)	1.8 (1.3)		2 (1.3)		1.2 (1.2)		**<0.0005**
**Household ownership**							
Owner/joint owner	872	26.08	745	24.0	127	52.3	**<0.0005**
Family house	601	17.98	515	16.6	86	35.4	
Rent free	252	7.54	249	8.0	3	1.2	
renting	1618	48.4	1591	51.3	27	11.1	
**Monthly household income (CFA) ****	**(*n* = 2723)**		**(*n* = 2534)**		**(*n* = 189)**		
<25 k (<42 USD)	237	8.7	188	7.4	49	25.9	**<0.0005**
26–50 k (43–83 USD)	922	33.9	847	33.4	75	39.7	
51–100 k (85–167 USD)	948	34.8	900	35.5	48	25.4	
101–200 k (168–333 USD)	428	15.7	417	16.5	11	5.8	
201–300 k (335–500 USD)	119	4.4	116	4.6	3	1.6	
301–500 k (502–834 USD)	54	2.0	51	2.0	3	1.6	
>500 k (>834 USD)	15	0.6	15	0.6	0	0	
**Method of payment**	**(*n* = 3178)**		**(*n* = 2935)**		**(*n* = 243)**		
Cash only	2313	72.8	2210	75.3	103	42.4	**<0.0005**
Cash and kind	280	8.8	193	6.6	87	35.8	
In kind	392	12.3	388	13.2	4	1.7	
Not paid	193	6.1	144	4.9	49	20.2	
**Access to transport**	**(*n* = 3340)**		**(*n* = 3098)**		**(*n* = 242)**		
Car	1024	30.7	980	31.6	44	18.2	**<0.0005**
	**(*n* = 3320)**		**(*n* = 3083)**		**(*n* = 237)**		
Pickup truck	241	7.3	230	7.5	11	4.6	0.107
	**(*n* = 3325)**		**(*n* = 3085)**		**(*n* = 240)**		
Motorbike	620	18.7	561	18.2	59	24.6	0.014
**Assets owned**	**(*n* = 3343)**						
Flush WC	1324	39.6	1285	41.5	39	16.1	**<0.0005**
Piped water	1370	41.0	1296	41.8	74	30.5	**0.001**
	**(*n* = 3336)**						
Livestock (*n* = 3336)	376	11.3	315	10.2	61	25.3	**<0.0005**

* *p*-values in bold show significance at the Bonferroni corrected level of 0.004. ** Extreme income poverty is defined at less than $1.9/day (approximately 1139 CFA).

**Table 4 ijerph-17-05874-t004:** Primary and secondary fuels used for cooking and degree of LPG use stratified by community.

	Total (*n* = 3342)		Botaland (*n* = 1059)		Batoke (*n* = 707)		Mile 4 (*n* = 1334)		Buea (*n* = 243)		*p* Value *
	No	%	No	%	No	%	No	%	No	%	
**Primary fuel**											
No cooking	17	0.5	3	0.3	6	0.9	6	0.5	2	0.8	
Electricity	11	0.3	4	0.4	3	0.4	3	0.2	1	0.4	
LPG	1883	56.3	713	67.3	364	51.6	768	57.6	38	15.6	
Piped gas	1	0.03	0	0	1	0.1	0	0	0	0	**<0.0005**
Kerosene	114	3.4	41	3.9	28	4.0	40	3.0	5	2.1	
Charcoal	51	1.5	10	0.9	5	0.7	35	2.6	1	0.4	
Wood	1205	36.1	279	26.4	285	40.4	445	33.4	196	80.7	
Sawdust	53	1.6	6	0.6	12	1.7	35	2.6	0	0	
Other	7	0.2	3	0.3	2	0.3	2	0.2	0	0	
**Secondary fuel**	**(*n* = 3325)**		**(*n* = 1056)**		**(*n* = 706)**		**(*n* = 1324)**		**(*n* = 239)**		
None	983	29.6	379	35.9	231	32.7	250	18.9	123	51.5	
Electricity	11	0.3	1	0.1	3	0.4	7	0.5	0	0	
LPG	554	16.7	119	11.3	121	17.1	279	21.1	35	14.6	
Piped gas	4	0.1	0	0	0	0	2	0.2	2	0.8	
Biogas	1	0.03	1	0.1	0	0	0	0	0	0	**<0.0005**
Kerosene	308	9.3	69	6.5	61	8.6	140	10.6	38	15.9	
Charcoal	428	12.9	115	10.9	40	5.7	265	20.0	8	3.4	
Wood	937	28.2	347	32.9	212	30.0	348	26.3	30	12.6	
Crops	1	0.03	0	0	1	0.1	0	0	0	0	
Sawdust	94	2.8	25	2.37	33	4.7	33	2.5	3	1.3	
Other	4	0.1	0	0	4	0.6	0	0	0	0	
**LPG use**	**(*n* = 3342)**		**(*n* = 1059)**		**(*n* = 706)**		**(*n* = 1334)**		**(*n* = 243)**		
None	913	27.3	230	21.7	222	31.4	291	21.8	170	70.0	**<0.0005**
In combination	1956	58.5	595	56.2	375	53.1	916	68.7	70	28.8	
Exclusive	473	14.2	234	22.1	109	15.4	127	9.5	3	1.2	

* *p*-values shown in bold show evidence of significance at the 5% level.

**Table 5 ijerph-17-05874-t005:** Perceptions of LPG in seven categories across the whole sample, and according to level of LPG use.

Perception	Total Sample **		No LPG (*n* = 913)		Some LPG (*n* = 1956)		Exclusive LPG (*n* = 473)		*p* Value *
	No	%	No	%	No	%	No	%	
**Speed of cooking**	(*n* = 2903)		(*n* = 496)		(*n* = 1936)		(*n* = 471)		
Very slow or slow	361	12.4	56	11.3	288	14.9	17	3.6	**<0.0005**
Very fast or fast	2542	87.6	440	88.7	1648	85.1	454	96.4	
**Safety**	(*n* = 2982)		(*n* = 573)		(*n* = 1937)		(*n* = 471)		
Very dangerous or dangerous	1917	64.3	432	75.4	1274	65.8	221	44.8	**<0.0005**
Very safe or safe	1065	35.7	141	24.6	663	34.2	260	55.2	
**Ability to cook most dishes**	(*n* = 2903)		(*n* = 492)		(*n* = 1938)		(*n* = 472)		
Very difficult or difficult	994	34.2	229	46.5	701	36.2	63	13.4	**<0.0005**
Easy or Okay	1909	65.8	263	53.5	1237	63.8	409	86.7	
**Affordability of refills**	(*n* = 2827)		(*n* = 462)		(*n* = 1900)		(*n* = 464)		
Very expensive or expensive	1958	69.3	339	73.4	1346	70.8	273	58.8	**<0.0005**
Cheap or Okay	869	30.7	123	26.6	554	29.2	191	41.2	
**Cleanliness**	(*n* = 2943)		(*n* = 523)		(*n* = 1947)		(*n* = 472)		
Very dirty or dirty	135	4.6	15	2.9	104	5.3	16	3.4	**0.022**
Very clean or clean	2808	95.4	508	97.1	1843	94.7	456	96.6	
**Ease of replacing cylinders**	(*n* = 2806)		(*n* = 404)		(*n* = 1932)		(*n* = 468)		
Very difficult or difficult	1087	38.7	169	41.7	795	41.2	123	26.3	**<0.0005**
Easy or okay	1719	61.3	236	58.3	1137	58.9	345	73.7	
**Availability**	(*n* = 2769)		(*n* = 362)		(*n* = 1937)		(*n* = 469)		
Very difficult/difficult to obtain	1048	37.9	156	43.1	782	40.4	109	23.2	**<0.0005**
Easy or okay to obtain	1721	62.2	206	56.9	1155	59.6	360	76.8	

* *p*-values shown in bold show evidence of significance at the 5% level. ** Total sample size for perception data was less than 3343 due to some households not answering all questions.

**Table 6 ijerph-17-05874-t006:** Likelihood of viewing LPG positively in seven categories, according to the main socio-demographic factors.

	Perception of LPG													
	Fast		Safe		Clean		Available		Cheap		Cooks Most Dishes		Easy to Replace Cylinder	
	*n*	%	*n*	%	*n*	%	*n*	%	*n*	%	*n*	%	*n*	%
**Sex**														
Male	1169	86.2	474	34.1	1305	94.7	690	52.0	361	26.8	861	63.5	744	56.2
Female	1373	88.8	591	37.2	1503	96.0	1031	71.5	508	34.3	1048	67.8	975	65.8
*p* value *	0.038		0.076		0.084		**<0.0005**		**<0.0005**		0.014		**<0.0005**	
**Age**														
13–35	1410	90.0	647	40.7	1516	96.0	1021	68.5	509	33.6	1122	71.8	975	64.7
36–60	966	83.5	370	30.8	1117	94.8	616	55.4	316	27.8	687	59.2	645	57.3
61+	166	92.2	48	25.0	175	94.6	84	50.6	44	25.0	100	55.9	99	57.2
*p* value *	**<0.0005**		**<0.0005**		0.323		**<0.0005**		0.001		**<0.0005**		**<0.0005**	
**Education**														
None	62	91.2	34	45.3	70	95.9	35	67.3	22	34.4	38	57.6	25	44.6
Primary	786	86.1	247	25.6	888	95.9	478	57.9	231	26.3	535	58.7	502	58.6
Secondary	1211	89.4	544	39.7	1309	95.7	842	63.6	429	32.3	922	68.2	822	61.9
University	483	85.0	240	42.0	541	93.9	366	64.4	187	33.5	414	72.1	370	65.4
*p* value *	0.016		**<0.0005**		0.297		0.023		0.007		**<0.0005**		0.004	
**Income**														
<50 k	824	88.4	328	33.4	910	95.6	533	61.8	265	29.0	579	62.0	550	61.9
51–100 k	742	87.1	331	37.9	830	95.6	544	65.9	250	29.9	593	69.9	516	62.4
101–200 k	370	89.8	169	41.1	392	94.7	242	60.5	133	33.4	299	72.6	238	59.5
201–300 k	103	88.0	46	39.3	105	89.7	61	52.1	50	43.5	76	65.5	64	55.7
301 K+	57	85.1	36	53.7	58	85.3	42	61.8	35	53.9	39	57.4	40	59.7
*p* value *	0.620		0.002		**<0.0005**		0.034		**<0.0005**		**<0.0005**		0.609	
**Rurality**														
Rural	135	83.9	36	20.0	154	93.9	56	40.0	29	18.2	74	45.1	59	39.9
Peri-urban	2407	87.8	1029	36.7	2654	95.5	1665	63.3	840	31.5	1835	67.0	1660	62.5
*p* value *	0.142		**<0.0005**		0.341		**<0.0005**		**<0.0005**		**<0.0005**		**<0.0005**	

* *p* values in bold show significance at the Bonferroni corrected level of 0.001.

**Table 7 ijerph-17-05874-t007:** Perceptions of LPG associated with extent of LPG use in peri-urban and rural households.

Perception	Exclusive LPG Use					Any LPG Use				
	No	%	OR	95% CI	*p* Value *	No	%	OR	95% CI	*p* Value *
**Speed**										
Slow	17	5.1	1			296	88.4	1		
Fast	451	18.7	4.31	2.62, 7.10	**<0.0005**	2040	84.6	0.73	0.52, 1.04	0.083
**Safety**										
Dangerous	210	11.8	1			1437	81.1	1		
Safe	258	25.1	2.49	2.04, 3.05	**<0.0005**	899	87.5	1.63	1.31, 2.03	**<0.0005**
**Cooks most dishes**										
Difficult	63	7.0	1			734	81.3	1		
Easy	406	22.1	3.79	2.87, 5.01	**<0.0005**	1604	87.4	1.60	1.29, 1.99	**<0.0005**
**Affordable (refills)**										
Expensive	270	14.8	1			1569	85.8	1		
Cheap	191	22.8	1.70	1.38, 2.09	**<0.0005**	724	86.3	1.04	0.82, 1.32	0.750
**Cleanliness**										
Dirty	16	12.8	1			115	92.0	1		
Clean	453	17.1	1.40	0.82, 2.39	0.214	2232	84.1	0.46	0.24, 0.89	0.020
**Replacing cylinders**										
Difficult	122	12.2	1			878	88.0	1		
Easy	343	20.7	1.87	1.50, 2.34	**<0.0005**	1451	87.5	0.95	0.75, 1.21	0.697
**Availability**										
Difficult	108	11.2	1			852	88.5	1		
Easy	358	21.5	2.17	1.72, 2.73	**<0.0005**	1484	89.1	1.07	0.83, 1.37	0.606

* p values in bold show significance at the Bonferroni corrected level of 0.004.

**Table 8 ijerph-17-05874-t008:** Demographic/household factors associated with any and exclusive LPG use in peri-urban households (univariable analysis).

	Exclusive Use of LPG (*n* = 470; 15.2%)					Any Use of LPG (*n* = 1883; 56.3%)				
	No	%	OR	95% CI	*p*-Value *	No	%	OR	95% CI	*p*-Value *
**Sex (household head)**										
Male	248	17.2	1			1148	79.5	1		
Female	222	13.4	0.75	0.61, 0.91	0.004	1208	73.0	0.70	0.59, 0.82	**<0.0005**
**Age (household head)**										
13–35	362	22.3	1			1313	80.9	1		
36–60	100	7.9	0.30	0.24, 0.38	**<0.0005**	924	73.2	0.64	0.54, 0.77	**<0.0005**
61+	8	3.8	0.14	0.07, 0.28	**<0.0005**	119	56.1	0.30	0.22, 0.41	**<0.0005**
**Education (household head)**										
None	2	2.1	1			36	37.1	1		
Primary	92	8.9	4.65	1.13, 19.17	0.034	629	61.0	2.64	1.72, 4.07	**<0.0005**
Secondary	249	17.7	10.21	2.50, 41.71	**0.001**	1170	83.2	8.36	5.41, 12.92	**<0.0005**
University	127	22.6	13.84	3.36, 56.92	**<0.0005**	521	92.5	21.01	12.52, 35.23	**<0.0005**
**Marital status (household head)**										
Married/partner	212	11.6	1			1401	77.7	1		
Single/Widow/ Divorced	258	19.9	1.87	1.53, 2.28	**<0.0005**	955	73.8	0.81	0.68, 0.95	0.012
**People resident**										
1–3 people	335	32.5	1			789	76.5	1		
4–6 people	112	7.7	0.17	0.14, 0.22	**<0.0005**	1122	76.6	1.01	0.83, 1.21	0.948
7+ people	23	3.8	0.08	0.05, 0.13	**<0.0005**	445	73.8	0.86	0.69, 1.09	0.216
**People per room**										
0–1.5	278	21.9	1			998	78.6	1		
1.6–2	122	15.4	0.65	0.51, 0.82	**<0.0005**	624	78.6	1.00	0.81, 1.24	0.997
2.1–14	70	6.8	0.26	0.20, 0.34	**<0.0005**	734	70.9	0.66	0.55, 0.80	**<0.0005**
**Household ownership**										
Owner/joint owner	60	8.1	1			546	73.3	1		
Not a house owner	410	17.4	2.41	1.81, 3.20	**<0.0005**	1810	76.9	1.21	1.00, 1.46	0.045
**Income (CFA)**										
<50 k	155	15.0	1			666	64.4	1		
51–100 k	141	15.7	1.06	0.82, 1.35	0.666	711	79.1	2.10	1.71, 2.57	**<0.0005**
101–200 k	73	17.5	1.20	0.89, 1.63	0.231	371	89.0	4.47	3.21, 6.23	**<0.0005**
201–300 k	26	22.4	1.64	1.03, 2.62	0.038	110	94.8	10.16	4.42, 23.33	**<0.0005**
301 K+	12	18.2	1.26	0.66, 2.41	0.482	64	97.0	17.73	4.32, 72.85	**<0.0005**
**Method of payment**										
Cash only	397	18.0	1			1795	81.3	1		
Not cash only	58	8.0	0.40	0.30, 0.53	**<0.0005**	448	61.8	0.37	0.31, 0.45	**<0.0005**
**Access to transport**										
Car	159	16.2	1.13	0.91, 1.39	0.262	837	85.5	2.34	1.91, 2.86	**<0.0005**
Truck	42	18.3	1.27	0.90, 1.81	0.178	174	75.7	0.98	0.72, 1.34	0.901
Motorbike	92	16.4	1.11	0.87, 1.43	0.398	425	75.8	0.99	0.80, 1.22	0.895
**Assets**										
Flush WC	224	17.4	1.35	1.10, 1.64	0.003	1172	91.2	5.52	4.45, 6.85	**<0.0005**
Piped water	214	16.5	1.20	0.98, 1.46	0.077	1145	88.4	3.71	3.05, 4.51	**<0.0005**
**Livestock**										
Owned	8	2.55	0.13	0.06, 0.27	**<0.0005**	222	70.7	0.74	0.57, 0.95	0.020
**Cooking location**										
Inside house	282	23.0	1			1104	90.2	1		
Separate building	161	16.8	0.67	0.54, 0.83	**<0.0005**	729	75.9	0.34	0.27, 0.44	**<0.0005**
Outside	27	3.0	0.10	0.07, 0.15	**<0.0005**	523	57.8	0.15	0.12, 0.19	**<0.0005**

* *p*-values in bold have significance at the Bonferroni corrected level of 0.002.

**Table 9 ijerph-17-05874-t009:** Socio-demographic factors associated with any and exclusive LPG use in peri-urban households (using two multivariable logistic regression models).

Characteristic	Any LPG Use (*n* = 1883) Model R-Square = 0.3433 Goodness-of-Fit (*p* = 0.3811)			Exclusive LPG Use (*n* = 470) Model R-Square = 0.2363 Goodness-of-Fit (*p* = 0.8128)		
	OR	95% CI	*p* Value *	OR	95% CI	*p* Value *
**Sex (household head)**						
Male	1					
Female	0.93	0.73, 1.18	0.537			
**Age (household head)**						
13–35	1			1		
36–60	0.69	0.54, 0.88	**0.003**	0.45	0.34, 0.59	**<0.0005**
61+	0.41	0.25, 0.65	**<0.0005**	0.24	0.10, 0.55	**0.001**
**Education (household head)**						
None	1			1		
Primary	2.03	1.10, 3.75	**0.024**	2.00	0.45, 9.00	0.365
Secondary	3.76	2.01, 7.03	**<0.0005**	2.98	0.67, 13.30	0.153
University	5.81	2.83, 11.94	**<0.0005**	3.99	0.86, 17.51	0.078
**Marital status (household head)**						
Married/partnership				1		
Single/Widowed/Divorced				1.41	1.12, 1.78	**0.004**
**People resident**						
1–3 people				1		
4–6 people				0.26	0.20, 0.34	**<0.0005**
7+ people				0.18	0.11, 0.30	**<0.0005**
**People per room**						
0–1.5	1			1		
1.6–2	1.02	0.76, 1.37	0.878	0.95	0.71, 1.27	0.716
2.1–14	0.94	0.73, 1.22	0.640	0.47	0.33, 0.65	**<0.0005**
**Household ownership** Owner/joint owner Not a house owner						
				1		
				0.97	0.68, 1.39	0.866
**Household income (CFA)**						
<50 k	1					
51–100 k	1.63	1.28, 2.09	**<0.0005**			
101–200 k	2.23	1.52, 3.29	**<0.0005**			
201–300 k	3.11	1.24, 7.79	**0.015**			
301 K+	3.61	0.83, 15.67	0.086			
**Method of payment**						
Paid in cash only	1			1		
Not paid exclusively in cash	0.70	0.55, 0.90	**0.005**	0.59	0.43, 0.83	**0.002**
**Assets owned**						
Flush WC	2.23	1.66, 3.13	**<0.0005**			
Piped water	1.62	1.21, 2.19	**0.001**			
**Access to transport**						
Car	1.42	1.08, 1.85	**0.011**			
**Livestock**						
Owned by household				0.36	0.17, 0.75	**0.007**
**Cooking location**						
Inside the house	1			1		
Separate building	0.35	0.26, 0.47	**<0.0005**	1.03	0.79, 1.33	0.843
Outside	0.21	0.16, 0.28	**<0.0005**	0.17	0.11, 0.26	**<0.0005**

* *p*-values in bold show evidence of significance at the 5% level.

**Table 10 ijerph-17-05874-t010:** Summary of the factors which significantly influence both LPG adoption and exclusive use.

Enablers		Barriers	
Any LPG Use	Exclusive LPG Use	Any LPG Use	Exclusive LPG Use
Rising level of education	Being single	Rising age	Rising age
Rising income	Income paid in cash	Cooking outside	Increasing household size (people resident)
Household assets			Overcrowding
Access to a car			land ownership
Payment in cash			Cooking outside
Can cook most meals	Opinion LPG is fast	Opinion LPG unsafe	Opinion LPG unsafe
	Can cook most meals		Refills are expensive
	Cylinders easy to replace		Opinion LPG refills are unavailable

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
