# Peer review of "Drivers of the Adoption and Exclusive Use of Clean Fuel for Cooking in Sub-Saharan Africa: Learnings and Policy Considerations from Cameroon"

_ijerph, 2020, doi:10.3390/ijerph17165874_

Round 1

Reviewer 1 Report

-The paper provides a good background, especially since it is very precise in its implications. However, it seems to me that it is essential to highlight the number of studies that have been developed in this topic from the 80s onwards. The authors indicate “To achieve these targets, much interest has previously been shown in burning biomass more cleanly using improved cookstoves”. But they don't mention anything else regarding this issue.

-Taking into account the extensive study that has been developed around this is essential to position this paper within the existing literature. Given the existing literature, and above all, given the two previous studies already developed around the specific use of LPG by LACE in 2016 and 2017, the contribution of this particular study is not clear.

-The methodology is not consistent with the objective indicated in the introduction. In the introduction it is mentioned that the objective is to investigate the socioeconomic factors associated with the use of LPG. However, in the materials and methodology the authors indicate that they do not seek to make a predictive model, only a model that shows correlations, and therefore they do not make goodness-of-fit analyzes. How is this explained?

-Although the authors explain in detail how the interviews were carried out, there is no further description of the logistic regression model used, they only say that they run this model with socio-economic and demographic variables. What are those variables?

-Furthermore, the authors mention the use of a second multivariate logistic regression, in which all the statistically significant factors from the previous regression are incorporated. Details of the multiple comparisons that where performed are not given: Against what other options is the use of LPG being tested? What assumptions did the authors make in building the model?

To give more clarity and coherence to the work, it would be good to incorporate the constructed regressions.

-The study does not present a novel topic and since the methodology is not clear, the novelty of the methodology is not clear either.

-Along with this, the results are similar to those of LACE. Did the authors have a different hypothesis? Did they want to test LACE’s results?

-Since there is little development and methodological explanation it is difficult to determine the significance of the content as a tool.

-The abstract should be more specific in the methods used, and the conclusion should present more information on the findings of the study.

Reviewer 2 Report

The paper addresses an important matter for understanding energy transition in Cameroon. It uses primary information from surveys and it has a sound methodological design.

Comments are intended to improve the theoretical framework and presentation of results.

The literature on the energy transition is not explicit. I think that the works developed by Sovacool(2011,2012) could improve the theoretical discussion. As well as the findings could contribute to this theoretical body.

For example, you could use this literature:

Sovacool, B. K. (2011). Conceptualizing urban household energy use: Climbing the “Energy Services Ladder”. Energy Policy, 39(3), 1659–1668. https://doi.org/10.1016/j.enpol.2010.12.041 Sovacool, B. K. (2012). The political economy of energy poverty: A review of key challenges. Energy for Sustainable Development, 16(3), 272–282. https://doi.org/10.1016/j.esd.2012.05.006   About the results.  

I have concerns about this:

The cross-sectional nature of the data meant that the aim of analysis was to explore the relationships between household factors and perceptions and LPG use rather than to create a predictive model. As such, the models were not assessed for goodness of fit.

I think the goodness of fit is not only useful for prediction. Actually,  it shows how well a model fits a set of observations. So, I would recommend assessing goodness of fit.   

I find that sometimes some very small sample sizes are used and described. My biggest concern is about table 3. Computed p-values could be distorted by these small categories. I recommend using put some categories in "other". In particular "piped gas" and "crops".   

About the presentation of the multivariate models, I find it weird that we do not have n size in the table. Also in table 8, you have 2 models with different co-variates. So, it is difficult to compare if you 1) do not have the goodness of fit; 2) independent and dependent variables are different, 3) you present OR. OR are difficult to compare between model. I suggest computing marginal effects.         

Reviewer 3 Report

I have carefully considered and read the manuscript entitled “Drivers of adoption and exclusive use of clean fuel for cooking in Sub-Saharan Africa: learnings and policy considerations from Cameroon” and have the following observations:

Household air pollution (HAP) caused by the combustion of solid fuels for cooking and heating is responsible for almost 5% of the global burden of disease. In response, the World Health Organisation (WHO) has recommended the urgent need to scale adoption of clean fuels, such as liquefied petroleum gas (LPG), in low and middle-income countries (LMIC). To understand the drivers of adoption and exclusive use of LPG for cooking, we analysed survey data from 3343 peri-urban and rural households in South West Cameroon. Most households reported LPG to be clean (95%) and efficient (88%), but many also perceived it to be expensive (69%) and unsafe (64%). Positive perceptions about LPG safety (OR=2.49, 95%CI=2.04, 3.05), cooking speed (OR=4.31, 95%CI=2.62, 7.10), affordability (OR=1.7, 95%CI=1.38, 2.09), availability (OR=2.17, 95%CI=1.72, 2.73), and its ability to cook most dishes (OR=3.79, 95%CI=2.87, 5.01), were significantly associated with exclusive LPG use. Socio-economic status (higher education) and household wealth (higher income) were also associated with a greater likelihood of LPG adoption. Effective strategies to raise awareness around safe use of LPG and interventions to address financial barriers are needed to scale wider adoption and sustained use of LPG for clean cooking, displacing reliance on polluting solid fuels.

Major comments and Suggestions for Authors: This paper is not with enough clarity about the aims and objectives, so please do it clarify for more understanding to meet the standard of readership of IJERPH. There are some errors in spelling, and some more clarifications, improvements in modeling techniques, acute conclusion, policy recommendations, and research limitations are needed for reconsidering that manuscript for publication in the International Journal of Environmental Research and Public Health.

In addition to the above, I have a few points for the authors to consider them before the publication of this work:

• The abstract should check thoroughly and compose it with a summative style without spelling mistakes and more focused on main impacting results and policy implications.

• Please highlight your contribution and novelty of this manuscript with accuracy in the introduction part before the arrangement description. Furthermore, the objectives of your study should elaborate clearly there in the introduction part.

• The literature and theoretical background and hypothesis construction should improve and add more relevant studies e.g. (latest) to grab and display more contemporary literature critically.

• Please update your literature with few latest studies if it suitable and improve hypothesis style as well.

• Recheck the references and their style is according to the journal requirements, and in-text and end-text should be the same and vice versa.

• In the Methodology part, please more detailed the results and discussion in presence of constructed hypothesis part for the actual output of this study for stakeholders and targeted policymakers.

• In the result and discussion section, some associated literature must be added to compare and contrast the key findings with the existing studies. Furthermore, Study limitations should be included in final conclusion part.

• The conclusion should be based on your results and discussion. So, do consider it and improve it based on the logic of your results.

• The conclusion does not properly describe as it was needful, hence please provide expansion in your conclusion-based estimations and provide some recommendations and policy implications more in detail.

• The acronyms should be defined at first appearance in the manuscript and then must be consistently used throughout the manuscript. Furthermore, the manuscript must be checked form typo errors and spelling checks.

Reviewer 4 Report

IJERPH-859036 Manuscript review

Overall comments – Thank you for the opportunity to review the manuscript entitled “Drivers of adoption and exclusive use of clean fuel for cooking in Sub-Saharan Africa: learning and policy considerations from Cameroon. The aim of this study was to investigate the barriers to adoption and exclusive use of LPG for cooking at Sub-Saharan Africa. The authors showed that most households perceived LPG to be expensive and unsafe, highlighting the need for policies to address lack of awareness and financial barriers to improve rates of adoption. This cross-sectional study has provided essential information for the effective implementation of clean fuels for cooking, which is needed to reduce household air pollution and for health protection. I am recommending this manuscript for publication after a minor revision. I recommend another proof-read of the manuscript, as there are some minor corrections needed. Please consider the comments below in your revision.

Specific comments:

Abstract – Great

Introduction – Very good.

Materials and methods –

  • I highly suggest organising this section with subheadings, data collection, participants, statistical analysis etc.

Results

  • With regards to household size, i.e. number of rooms, Table 2 and Table 9 show that increasing household size is a barrier to exclusive LPG use. However, it is not mentioned here is the results.

Discussion

  • You have included 3 peri-urban communities and only 1 urban community, and this disparity is reflected in the difference in sample size. There are over 10x the participants in the peri-urban vs the rural cohorts. I recommend the authors comment on the impact of this in the limitations section.
  • Again, with regards to household size, the authors discuss some of the perceptions around LPG use in order to identify potential barriers. One perception I believe to be over looked is necessity, do the participants believe that LPG use is necessary in their living environment? For instance, if the urban population tend to live in a larger house, in a potentially less populated area with less outdoor air pollution, the perceived impact of HAP may be different to a peri-urban population, hence this population may be less encouraged to make the changes due the inconvenience of inaccessibility and the financial burden of LPG if they don’t believe the change is necessary. This is just an idea. But I highly recommend the authors consider this finding and comment on it in the discussion as a barrier for LPG use, as determined in the findings of the study. Also note, that household size in mentioned in the conclusion, but not in the results or discussion.

Round 2

Reviewer 1 Report

  1. We would like to thank the reviewer for their feedback. We have added reference to a systematic review and meta-analysis synthesising the evidence of cleaner cooking interventions of household air pollution (lines 46-51). We have also provided a narrative based on previous studies of drivers of adoption and use of clean fuels/ technologies (lines 86-102).

 With the new references, the previous interest in studying strategies that promote a cleaner use of biomass as a residential fuel is much clearer.

  1. We have added text in the introduction to clarify that this paper makes use of a large, unique database, including perceptions of LPG as a cooking fuel, and builds on the mixed-methods research previously published for LACE (lines 103-108).

The answer to this comment is still not clear to me. I understand that you are using the same databases that LACE used, the difference is that you joined them. If so, the novelty is not the use of this information.

I would understand the novelty in the use of these databases, in the context of a work that seeks to answer a different question than the LACE previous work. But, from what you explained I understand that LACE (2016 and 2017) collects this information with the same objective.

Could you add more information on the novelty of the study beyond joining both databases?

  1. We take the reviewer’s point (here and below) that by developing logistic regression models with LPG use as a dependent variable and the household characteristics etc. as independent variables, the results can be considered as going beyond the conservative description of ‘association’. We have therefore included descriptions of the models including statistical goodness-of-fit for the final multi-variable model (Hosmer-Lemeshow test). The statistical analysis section has been rewritten (lines 141- 159).
  2. We have now clarified the dependent (LPG use) and independent variables (socio- demographic, wealth, individual and household characteristics and the fuel perception variables) for the logistic regression model under statistical analysis (lines 141-159).

The information that has been added makes it much clearer now and responds to comments 3 and 4.

  1. We have clarified the dependent and independent variables for the regression models in the statistical analysis section (as described above). Univariate logistic regression was used to identify which of the independent variables were potential ‘predictors’ of the dependent variable (LPG use). The dependent variable was classified in two ways (i) any LPG use with reference group of no-LPG use and (ii) exclusive LPG use with reference group of secondary use of LPG or no use. The predictive properties of the independent variables were assessed statistically (corrected p-value using the Bonferroni correction) and interpreted through the odds ratios (and 95% confidence intervals). The final multivariable model included only those factors that achieved statistical significance through univariate analysis – as such the final model was assumption free.

The added information makes it much clearer now.

  1. We present the regression analysis outputs in Tables 6, 7 and 8. For Table 8 (multivariable regression analysis for (i) Any use of LPG and (ii) Exclusive use of LPG), we have presented the goodness of fit estimation (R2 and GOF p-values). We have also discussed this in the results when interpreting the multivariable models (lines 287-294).

The tables provide information that allow a much better understanding of the methodology.

  1. We hope the analytical methods are now clearer explaining how the analysis and modelling has been conducted and detailing the dependent and independent variables. The novelty of the research is the large robust dataset summarising key predictors of LPG use (including perceptions around convenience and safety) that coincide with publication of a national strategy to scale adoption of clean cooking with LPG in Cameroon.

Analytical methods make it much clearer that analysis is well supported. However, I still do not understand why the authors indicate that the novelty is the database, if this database has already been used for two previous studies on the same subject, and with the same approach.

  1. Whilst the previous LACE study looked at some of the same characteristics in relation to LPG use, this study is built on a considerably larger dataset from a larger geographical region and enquiring about perceptions of cooking with LPG that impact on its use. This robust and comprehensive analysis provides important community- based evidence for stakeholders in Cameroon developing policy alongside the National LPG Masterplan. The novel aspects of the current study are made clearer in the introduction (lines 103-108).
  2. The questionnaires used for the quantitative surveys were developed using existing pre-validated questions (e.g. WHO questions set for fuel use) and new questions derived through expert consultation and subsequent piloting (e.g. fuel preference questions). We have added a section on the development of the questionnaire in the methods (lines 128-131). The tools are now being used to conduct extensive quantitative survey-based work in Ghana, Kenya and Francophone Cameroon as part of clean cooking research for an NIHR Global Health Research Group (CLEAN-Air Africa).

In the Materials and Methods section (subsections 2.1 and 2.2) a description of the LACE databases from 2016 and 2017 is presented. Then, the new questions derived from consultations with experts and pilots are presented; but this is explained within the context of the LACE database.

If the novelty is the use of a single, much larger database, the information that was added to the LACE databases should be explained. Joining both bases is not a novelty in itself. But in that case, the novelty should be related to the methodology.

Further information on this issue could be added: What specific information is added? How and when was this new information collected? How was that information incorporated?

  1. A sentence has been added in the abstract to describe the tool used in the survey (lines 20-22). The conclusion has been expanded to present more information on the findings of the study (lines 431-446).

The added information makes it much clearer now.
